# Theoretical Limits of Star Sensor Accuracy [note 1]

**DOI:** 10.3390/s19245355

**Published:** 2019-12-05

**Authors:** Marcio A. A. Fialho, Daniele Mortari

**Affiliations:** 1Divisão de Eletrônica Aeroespacial (DIDEA), National Institute for Space Research (INPE), Av. dos Astronautas, 1758, São José dos Campos, SP 12227, Brazil; 2Aerospace Engineering, Texas A&M University, College Station, TX 77843, USA; mortari@tamu.edu

**Keywords:** star sensors, stellar distribution, photometry, astrometry, star catalogs, fundamental limits

## Abstract

To achieve mass, power and cost reduction, there is a trend to reduce the volume of many instruments aboard spacecraft, especially for small spacecraft (cubesats or nanosats) with very limited mass, volume and power budgets. With the current trend of miniaturizing spacecraft instruments one could naturally ask if is there a physical limit to this process for star sensors. This paper shows that there is a fundamental limit on star sensor accuracy, which depends on stellar distribution, star sensor dimensions and exposure time. An estimate of this limit is given for our location in the galaxy.

## 1. Introduction

Much progress in a variety of fields in science and technology could be accomplished thanks to the miniaturization obtained in microelectronics in the recent decades. One of the most remarkable examples is the prediction by Gordon Moore that the computational power would increase exponentially, an empirical observation that became known as Moore’s law [1,2,3,4]. Yet, this rate of improvement is not expected to last forever. Eventually a fundamental limit will be reached when the size of transistors reaches atomic scales. Likewise, in other fields of science and technology, fundamental limits to miniaturization and performance improvements are often found. For instance, in the field of telecommunications, there is a theoretical minimum amount of energy that must be spent to transmit a bit in a digital message from one point to another within a given time interval, this quantity being closely related to Planck’s constant [5]. Hence, it is natural to ponder whether there is a fundamental limit to the accuracy attainable by star sensors, given constraints such as the *volume* in space it occupies, the length of *time* available for observations, and the distribution and brightness of stars around it.

An attitude sensor is any instrument used aboard spacecraft to provide data to estimate its orientation in space (attitude). Many spacecraft need to have an accurate knowledge of their attitude in order to accomplish their mission goals (e.g., point cameras/telescopes, communication antennas, thrusters, etc.). In order to do so, they use a variety of attitude sensors, such as sun and horizon sensors, magnetometers and star sensors, which can be further classified into *star scanners*, used in spinning spacecraft and *star trackers*, usually employed in three-axis stabilized spacecraft [6,7]. Star trackers (STRs) are among the most accurate attitude sensors available for spacecraft use, by providing absolute triaxial attitude measurements with errors typically in the order of few arc-seconds or less [8,9]. These sensors are in essence computerized optical cameras with the appropriate software for star extraction, star identification, and attitude determination. Reference [10] provides a good explanation of how star trackers work.

The purpose of this study is to present an estimate for the ultimate limits for attitude determination from stars, imposed by fundamental laws of Physics, that is, limits that cannot be overcome by technology improvements, for our location in our galaxy. These estimates are useful as a basis for assessing real star sensors as to their potential for improvements through technology advancements. We will not discuss in this work practical limitations faced by existing, real world star sensors, such as readout noise, non-ideal point spread function (PSF) in centroiding and distortions introduced by the optics, since these limitations have already been well covered by the existing literature [11,12,13].

This paper is a revised and extended version of a work previously published as a doctoral thesis chapter [14], being organized as follows: Section 2 describes the methodology used, Section 3 presents and discusses the results, and Section 4 concludes this paper.

## 2. Methodology and Model Description

The star sensor model analyzed in this work is an ideal spherical star sensor, capable of measuring the direction and energy of every photon incident on its surface. This ideal star sensor is able to observe stars from any direction, that is, it has a field of view of 4πsr. The knowledge on the incoming direction of photons in this model is limited only by diffraction at the star sensor aperture, assumed to be circular with the same radius of the star sensor itself. In other words, it is assumed that the star sensor aperture is given by the projection of the star sensor body on a plane perpendicular to the direction of incoming photons. Figure 1 provides a sketch of the star sensor model adopted in this work. In this model, the accuracy of the centroids of each star is limited only by diffraction and shot noise. These effects depend only on the star sensor aperture, stellar spectra, and integration time (exposure time). This ideal star sensor is completely black, as it absorbs every photon impinging on it. Section 2.1 and Section 2.2 provide more details on the assumptions adopted in this model.

### 2.1. Basic Assumptions

The following basic assumptions were made:

**Assumption** **1.**
*The star sensor has a spherical shape with a diameter D.*


**Assumption** **2.**
*It is able to detect every photon of stellar origin impinging on its surface.*


**Assumption** **3.**
*It is capable of registering the incoming direction and energy of every detected photon with an accuracy limited only by Heisenberg’s Uncertainty Principle.*


**Assumption** **4.**
*Only photons detected during a period of length t—the exposure or integration time—are considered for attitude determination.*


**Assumption** **5.**
*It is at absolute zero temperature.*


**Assumption** **6.**
*The coordinates of the stars in an inertial reference frame with origin in the star sensor are known with absolute precision.*


**Assumption** **7.**
*An unbiased optimal estimator is used to determine the star sensor attitude, and computations are performed with infinite precision.*


**Assumption** **8.**
*Measurements obtained with this ideal star sensor are not merged with external measurements.*


Assumption 2 implies that the star sensor field of view is 4πsr, in other words, it is capable of observing the whole celestial sphere simultaneously, a fact that coupled to its spherical shape, implies that the accuracy of this ideal star sensor does not depend on its attitude.

Assumption 3 and the fact that every photon is detected implies that the optics are ideal: 100% transmission, with no defocusing and blurring, except for the blurring dictated by diffraction.

Assumption 5 means there is no noise of thermal origin within the star sensor.

Assumptions 4 and 8 limit the number of photons that will be observed by the ideal star sensor. If exposure time were not constrained, it would be possible to get attitude measurement uncertainty as low as desired, just by increasing the exposure time. In addition, this model does not consider the possibility of combining current observations with previous observations to improve accuracy. Assumption 4 also implies that the star sensor is able to measure just photons and no other particles. The basis for this assumption is the fact that the only other particle known to science that could, perhaps, convey better the positions of stars are neutrinos emitted at their core. However, these particles interact so weakly with ordinary matter that their detection in star sensors is currently impossible and may never become a reality [15].

Assumption 6 implies that the star catalog is perfect and that all corrections needed to bring the coordinates, brightness and colors from the star catalog reference frame origin to the star sensor location (corrections for stellar aberration, parallax, and redshift/blueshift) are performed with no errors.

Assumption 8 expresses the goal of obtaining a lower bound on attitude error for a single star sensor used in isolation. If measurements from multiple sensors were permitted to be merged, a significant improvement in attitude measurement accuracy would become possible. For example, by interferometrically combining measurements from a small number of star sensors mounted in a rigid structure and separated by a distance much greater than their diameters, it would be possible to improve attitude determination by many orders of magnitude in comparison to the theoretical estimate presented in this work, with attitude uncertainty being roughly inversely proportional to the distance between them [16,17].

### 2.2. Simplifying Assumptions

In addition to the previous assumptions, to make this study feasible, the following additional assumptions were also made:

**Assumption** **9.**
*The whole Universe is assumed to be composed only by stars.*


**Assumption** **10.**
*Stars are considered as polychromatic point sources of light.*


**Assumption** **11.**
*Stellar spectra are approximated by the spectra of black bodies that best match the cataloged star intensity given by star catalogs adopted here.*


**Assumption** **12.**
*All Solar System bodies (including the Sun) are disregarded.*


**Assumption** **13.**
*Stellar proper motion is disregarded.*


**Assumption** **14.**
*The star sensor is not rotating.*


**Assumption** **15.**
*It is assumed that each detected photon can be univocally associated with the star from where it originated.*


**Assumption** **16.**
*Stars are considered to be at an infinite distance.*


**Assumption** **17.**
*There is no obstruction from spacecraft structures or nearby bodies.*


Simplifying Assumption 9 means that we are not considering as additional sources of attitude information extended bodies, such as interstellar clouds, given that these sources are difficult to precisely model and would hardly significantly increase our attitude knowledge. However, it does not necessarily mean that all non-stellar pointlike sources will be excluded from analysis. This means simply that any non-stellar pointlike source present in star catalogs, such as some quasars and some distant galaxies, will be treated as if they were stars.

Regarding simplifying Assumption 11, ideally, the actual spectra of stars should be used, at least for the brightest stars, something to be attempted in future works. Section 3.2 discusses the adequacy of this approximation.

About simplifying Assumption 14, had the star sensor been rotating, but with knowledge of the precise instant each photon was detected and a very accurate knowledge of its own angular velocity vector, it would be possible to compute the incoming direction of every photon in a non-rotating reference frame attached to the star sensor, thus reducing the problem of attitude determination of a spinning star sensor to the problem of attitude determination of a non-rotating star sensor.

Regarding simplifying Assumption 17, had there been any obstruction in the field of view of the ideal star sensor (such as obstruction from nearby bodies or obstruction by spacecraft structures), its accuracy would necessarily be worse, since the obstructed stars would no longer contribute to the attitude information gathered by the star sensor. Therefore, we assume that it has an unrestricted view of the whole celestial sphere. In Section 3.4 we investigate the effects of reductions in the field of view.

Our computations disregard the Sun and other Solar System objects as additional references for attitude determination, since these sources are difficult to model accurately. Also, the Sun being many orders of magnitude closer and brighter than the other stars, from our vantage point in the Universe, it is too bright to be directly observed by most, if not all, star sensors. However, an attitude sensor in a distant future which is able to use and model very accurately the Sun and a neighboring planetary body as additional attitude references, could, perhaps, overcome the estimates on the lower bound of attitude uncertainty computed in this work. This is a topic to be better investigated in the future.

### 2.3. Model Description

Figure 2 presents a flowchart for the model used in this work. Basically, for each star in the selected star catalog, an estimate for the lower bound on centroiding uncertainty is computed and these estimates are used together with the unit vectors that represent the stars in the star catalog reference frame to determine the lower bound on attitude determination uncertainty (box at the lower right corner).

Unfortunately, no star catalog is complete. Therefore, any estimate obtained from an existing star catalog will be incomplete, since the missing stars in that star catalog still can contribute to attitude knowledge if they are observed by the star sensor, no matter how far or dim they are. To work around this limitation, we plot the relation of attitude knowledge upper bound with star catalog size for a number of publicly available star catalogs and extrapolate that to the estimated number of stars in our galaxy, plus some margin, to account for extragalactic sources, as described in Section 3.3. In the following sections, a more detailed description of the model used is given.

### 2.4. Black Body Model for Stars

In the model adopted in this work, the spectrum of each star is considered to be the spectrum of an equivalent spherical black body, diluted by a non-dimensional geometric factor *C* arising from its distance to the star sensor. Given that the spectral exitance of a black body is uniquely determined by its temperature, only two parameters are needed in this model to determine the spectral distribution and intensity of the electromagnetic radiation received by the star sensor from each star, the temperature *T* and the dilution factor *C*. Mathematically:(1)$$E_{e,\lambda ,i}\left(\lambda \right)=C_{i}\cdot M_{e,\lambda }\left(T_{i},\lambda \right)$$
where:$E_{e,\lambda ,i}\left(\lambda \right)$ is the spectral irradiance received from star *i* by a surface located at the same place of the star sensor and perpendicular to incoming rays, evaluated at wavelength λ;$C_{i}$ is the geometric dilution factor for star *i*;$T_{i}$ is the temperature of the black body that represents star *i*;$M_{e,\lambda }\left(T_{i},\lambda \right)$ is the spectral exitance of the surface of the equivalent black body, at wavelength λ.

In this equation, both $E_{e,\lambda ,i}\left(\lambda \right)$ and $M_{e,\lambda }\left(T_{i},\lambda \right)$ are given in unit of power per unit of area and per unit of wavelength (e.g., W/m${}^{2}$/nm).

To uniquely determine these two parameters (*T* and *C*) for each star, at least two samples of their flux taken at different wavelengths or at different spectral bands are needed. The following sections describe how *T* and *C* are derived for each star from Hipparcos catalog data [18], using the cataloged $m_{V}$ magnitudes and B−V color indexes. A similar procedure is performed with data from Hipparcos using the V−I color indexes and data from other star catalogs.

### 2.5. Black Body Temperatures from B−V Color Indexes

Taking as an example data from the Hipparcos catalog, the spectra of stars is taken as the spectra of black bodies with intensities adjusted so that the integrated spectra over the Johnson’s *B* and *V* bands [19,20] match simultaneously the flux at these bands derived from catalog data. To determine equivalent black bodies temperatures for stars in the Hipparcos catalog, an empirical relation is established in this section, linking the B−V color indexes given in the Hipparcos catalog with black body temperatures.

The spectral exitance at wavelength λ of a black body at a temperature *T* can be computed as follows [21]:(2)$$M_{e,\lambda }\left(T,\lambda \right)=\frac{2\pi hc^{2}}{\lambda^{5}}\frac{1}{e^{\frac{hc}{\lambda kT}}-1},$$
where *h* is the Planck’s constant, *c* is the speed of light in vacuum and *k* is the Boltzmann constant. The spectral exitance will have units of power per unit area per unit wavelength ($\left[W\cdot m^{-2}\cdot m^{-1}\right]$ in SI units). Numerical values of *h*, *c* and *k* used in computations were those adopted in the 2019 redefinition of the SI base units [22].

By integrating the product of the spectral exitance of a black body with the Johnson’s *B* and *V* bands energy responses it is possible to obtain the black-body fluxes in the *B* and *V* bands at its surface. This procedure is described in detail by Bessell in Reference [19], Section 1.6—Synthetic Photometry:(3)$$\varphi_{BB,B}\left(T\right)=\int_{\lambda =0}^{\infty }M_{e,\lambda }\left(T,\lambda \right)R_{B}\left(\lambda \right)d\lambda$$
(4)$$\varphi_{BB,V}\left(T\right)=\int_{\lambda =0}^{\infty }M_{e,\lambda }\left(T,\lambda \right)R_{V}\left(\lambda \right)d\lambda ,$$
where $\varphi_{BB,B}\left(T\right)$ = flux at the surface of a black body at temperature *T* in the Johnson’s *B* band and $R_{B}\left(\lambda \right)$ = spectral energy response function of the Johnson’s *B* band. Analogously, $\varphi_{BB,V}\left(T\right)$ and $R_{V}\left(\lambda \right)$ are quantities related to the Johnson’s *V* band.

The $R_{B}\left(\lambda \right)$ and $R_{V}\left(\lambda \right)$ response functions were obtained by converting the tabulated values recommended by Bessell (Table 1 on page 146 of Reference [20]) from normalized photonic responses to normalized energy responses and interpolating the resulting values. The energy response functions adopted in this work are shown in Figure 3.

The conversion from normalized photonic response to normalized energy response was done by multiplying the photonic response by the wavelength and renormalizing the results (Equation (A9) in Reference [20]). The explanation for this procedure is given in Section A2 in the appendix of Reference [20], on page 153. The method of interpolation used was a “shape-preserving piecewise cubic interpolation”, provided by the MATLAB/GNU Octave function interp1 with method “pchip”. Computations were performed in MATLAB R2015b with the script plot_BV_BB_script.m from the .zip archive which supplements this work (see “Appendix A” on page 20).

From the fluxes in the *B* and *V* bands, the magnitudes in these bands can be computed:(5)$$m_{BB,B}\left(T\right)=-2.5\;\log_{10}\left(\varphi_{BB,B}\left(T\right)/\varphi_{REF,B}\right)$$
(6)$$m_{BB,V}\left(T\right)=-2.5\;\log_{10}\left(\varphi_{BB,V}\left(T\right)/\varphi_{REF,V}\right).$$

These equations give the apparent magnitudes in the *B* and *V* spectral bands of a spherical black body for an observer situated just above its surface looking down towards its center. $\varphi_{REF,B}$ and $\varphi_{REF,V}$ are the reference fluxes that define the zero points of the magnitude scales in these bands, having being obtained by numerically integrating the spectrum of Vega (α-Lyr) multiplied by the band responses, and adjusting their values such that the computed *B* and *V* magnitudes of Vega matches those in the star catalog ($m_{\mathrm{Vega},B}=0.029$ and $m_{\mathrm{Vega},V}=0.030$ in Hipparcos). Mathematically:(7)$$\varphi_{REF,B}=10^{0.4m_{Vega,B}}\int_{\lambda =0}^{\infty }E_{Vega}\left(\lambda \right)R_{B}\left(\lambda \right)d\lambda$$
(8)$$\varphi_{REF,V}=10^{0.4m_{Vega,V}}\int_{\lambda =0}^{\infty }E_{Vega}\left(\lambda \right)R_{V}\left(\lambda \right)d\lambda ,$$
where $E_{Vega}\left(\lambda \right)$ is the spectral irradiance from Vega measured at the top of Earth’s atmosphere. The spectrum of Vega used in Equations (Equation 7) and (Equation 8) was obtained from file alpha_lyr_stis_008.fits from the CALSPEC database [23], which at the time of this writing was available at http://www.stsci.edu/hst/instrumentation/reference-data-for-calibration-and-tools/astronomical-catalogs/calspec.

Figure 4 shows the apparent magnitudes of black bodies versus temperature in the Johnson’s *B* and *V* bands for an observer located at their surface. In this plot, brighter sources (more negative magnitudes) are at the top. Note that the magnitude scale used in astronomy is reversed, with smaller magnitudes meaning brighter sources. The magnitudes are said to be apparent because they depend on the observer location, contrasting to stellar absolute magnitudes which are magnitudes of a star as seen from a standardized distance [24].

The difference between the *B* and *V* magnitudes of a celestial body is its B−V color index. Figure 5 presents the relation between the B−V color index and temperature for black-bodies. The plot to the right relates the B−V color index with the multiplicative inverse of its temperature. Note that this curve is much more linear than the direct relation between temperature and B−V color index. Therefore, to get equivalent black-body temperatures for stars in the catalog, we use the 1/*T* versus B−V curve for interpolation. To avoid temperature estimates with large errors from appearing, the B−V color indexes in the Hipparcos catalog are clamped into the interval [−0.2357,+2.7028] before conversion. These limits correspond to black-body temperatures of 30,000 K and 2000 K, respectively. Most stars have effective temperatures in that range.

It should be noted that for many stars, the temperature *T* used in our model will not be equal to the effective temperature of the star but will usually be smaller. This is caused by interstellar reddening—selective absorption by dust in the intervening light path from that star to the star sensor. Likewise, the constant *C* will also have a different value.

### 2.6. Determination of the Geometric Dilution Factor *C* from Hipparcos Data

From temperature *T*, the equivalent black body’s visual magnitude at its surface ($m_{BB,V,surface}$) is determined by interpolating the solid black curve in Figure 4. The dilution factor *C* is then obtained by comparing this magnitude with the cataloged visual magnitude ($m_{V}$) in the Hipparcos catalog, using the following equation:(9)$$C_{i}=10^{0.4\cdot (m_{BB,V,surface,i}-m_{V,i})}$$

The geometric dilution factor *C* will typically be between $10^{-20}$ and $10^{-14}$ for stars in the Hipparcos catalog. In this equation, the subscript *i* indicates that the values refer to star *i*.

### 2.7. Number of Photons Detected Per Unit Wavelength

This section derives equations for the number of photons that will be detected, per wavelength, by the idealized star sensor used in this model, for a given exposure time *t* and a given star sensor diameter *D*, also assumed to be equal to its aperture diameter. The energy of each photon is related to its frequency ν by the following equation:(10)$$E_{ph}=h\nu =\frac{hc}{\lambda }.$$

Dividing the spectral irradiance at the location of the star sensor due to the black-body equivalent of star *i* (Equations (Equation 1) and (Equation 2)) by the energy of a photon of wavelength λ, the following expression for the spectral photon flux density received by the star sensor from the equivalent of star *i* is obtained:(11)$$\varphi_{ph,\lambda ,i}=C_{i}\frac{2\pi c}{\lambda^{4}}\frac{1}{e^{\frac{hc}{\lambda kT_{i}}}-1}.$$

This flux density has units of photons per unit of time per unit of area per unit of wavelength. Multiplying this by the star sensor’s cross section area $A=\pi D^{2}/4$ and by the integration time *t* we obtain:(12)$$n_{ph,\lambda ,i}=C_{i}\cdot t\cdot \frac{\pi^{2}D^{2}c}{2\cdot \lambda^{4}}\frac{1}{e^{\frac{hc}{\lambda kT_{i}}}-1},$$
which is the number of photons from star *i* black body equivalent being collected by the star sensor, per unit wavelength.

### 2.8. Diffraction and Photon Noise

Diffraction and optics blurring set the format of the point spread function (PSF) of stellar image. In an ideal star sensor, there is no optical blurring, except for that set by diffraction. Therefore, for the star sensor model adopted in this work, the PSF function will be the diffraction pattern given by a circular aperture of diameter *D* contained in a plane perpendicular to the incoming direction of photons. This is the well known Airy pattern, consisting of a center disk with a series of concentric rings [25].

If the description of Nature given by Classical Mechanics were correct, it would be possible, at least in theory, to measure the intensity of the electromagnetic fields at the detector plane of an ideal equivalent system composed by ideal lens and ideal detector with no error, from where the true, error free direction of the incoming light rays would be obtained. However, the fact that light is discretized in photons limits the amount of information that can be obtained, regardless of the system considered. Instead of precisely defining the intensity of the electromagnetic fields at each point in the detector (as thought by 19th century physicists), the PSF defines the probability density function that a photon coming from a point source at infinity will be detected in a particular location on the detector. Given that the number of photons detected is finite and these arrive at random positions following the PSF, the centroid estimate for each observed star will be noisy, even for an ideal system. The lower bound for the resulting uncertainty in centroiding can be obtained by many different methods [26,27,28]. Section 4.1.1 in Reference [26] provides a detailed explanation, targeted at telescopes and interferometers. In the next section we show how this derivation can be extended for polychromatic sources (black-bodies).

### 2.9. Lower Bound on Centroiding Error for Single Stars

Heisenberg’s uncertainty principle sets a fundamental limit for centroiding, and this limit assumes the following form for monochromatic light of wavelength λ [26,27,28]:(13)$$\sigma_{xc}⩾\frac{\lambda }{4\pi \Delta x\sqrt{N}}$$
where:$\sigma_{xc}$ = angular centroiding uncertainty along an axis *x* perpendicular to the direction of incoming photons, in radians;$\Delta x=\sqrt{\int {\left(x-\bar{x}\right)}^{2}\;dS\;/\int dS}$ is the root mean square extension of the star sensor aperture (entrance pupil) along the *x* axis, being $\bar{x}=\int x\;dS\;/\int dS$ the position in *x* of the aperture geometric center; and*N* = number of photons detected.

For circular apertures of diameter *D*, Δx=D/4. Substituting this into Equation (Equation 13) the following expression for the reciprocal of the lower bound of variance of centroiding error (the Fisher information *F*) over a circular aperture of diameter *D*, for monochromatic sources of light, is obtained:(14)$$\frac{1}{\sigma_{xc}^{2}}⩽\frac{1}{\sigma_{\min}^{2}}≜F_{N,\mathrm{mono}}=\frac{\pi^{2}D^{2}N}{\lambda^{2}}.$$

Since stars are incoherent sources of light, the detection of a given photon is not correlated with the detection of another photon from the same star. This means that the number of detected photons from a given star will follow a Poisson distribution with parameter ι, being ι the expected number of detected photons (we are using the Greek letter ι instead of the more common λ for the Poisson distribution parameter to avoid confusion with λ used for wavelength). This parameter can be obtained by integrating Equation (Equation 12). For large values of ι, the Poisson distribution narrows down in comparison to the value of ι. This means that when the expected number of detected photons is significantly large, the true value of the lower bound of centroiding accuracy will be very close to the value predicted by Equation (Equation 14) if we substitute *N* by ι. Numerical tests have shown, assuming that the centroiding error for exactly *N* detected photons follows a Gaussian distribution with a standard deviation given by Equation (Equation 13), that the error between the actual centroiding error and the value estimated by Equation (Equation 14) using ι in place of *N* will be smaller than 23% for N≥1, 6.4% for N≥10 and 0.51% for N≥100. It is true that the actual probability density function for centroiding error along one axis will not be exactly Gaussian, especially for a low number of detected photons, but a Gaussian distribution provides a good approximation even when only one photon is detected.

Another consequence of the fact that the detection of a given photon is not correlated with the detection of another photon from the same star is that the centroiding error of a centroid computed using photons in the wavelength interval $\left[\lambda_{1},\lambda_{2}\right]$ is independent on the centroiding error using photons in the wavelength interval $\left[\lambda_{3},\lambda_{4}\right]$ when these intervals do not overlap ($\lambda_{2}<\lambda_{3}$ or $\lambda_{4}<\lambda_{1}$). Therefore, we can consider each wavelength interval individually and then merge the centroid estimates for each wavelength.

For the discrete case of having *n* independent unbiased estimates of the same physical variable (e.g., the *x* coordinate of a star centroid), each having a variance $\sigma_{i}^{2}$, the best estimate for that variable is obtained by summing these estimates using the reciprocal of their variances as weights [27,29]. In that case, the variance of this optimal estimate will be given by:(15)$$\sigma_{T}^{2}={\left(\sum_{i=1}^{n}\sigma_{i}^{-2}\right)}^{-1},$$
where $\sigma_{T}^{2}$ = total variance in the estimate of a scalar physical variable obtained by merging *n* independent measurements and $\sigma_{i}^{2}$ = variance of each individual measurement *i*. Since the spectra of black bodies is continuous, the following adaptation of Equation (Equation 15) is used to compute centroid estimates for black bodies:(16)$$\frac{1}{\sigma_{xc}^{2}}=\int_{\lambda =0}^{\infty }\frac{d(\sigma^{-2})}{d\lambda }d\lambda .$$

The contribution from each wavelength to the knowledge of the centroid position can be obtained from Equation (Equation 14) by replacing *N* with $n_{ph,\lambda ,i}d\lambda$, where $n_{ph,\lambda ,i}=dN_{ph,i}/d\lambda$ is the derivative with wavelength of the number of photons from star *i* entering the star sensor aperture within an integration time of *t*, as given by Equation (Equation 12) from Section 2.7. Hence, for each star *i*, the wavelength derivative of the maximum knowledge physically attainable of its centroid position (derivative of its centroiding Fisher information) is given by:(17)$$\frac{dF}{d\lambda }=\frac{d\left(\sigma_{\min}^{-2}\right)}{d\lambda }=\frac{\pi^{2}D^{2}}{\lambda^{2}}\;n_{ph,\lambda }.$$

Here we have dropped the subscript *i* to simplify notation. Plugging Equation (Equation 12) into Equation (Equation 17) yields:(18)$$\frac{dF}{d\lambda }=C\cdot t\cdot \frac{\pi^{4}D^{4}c}{2\lambda^{6}}\frac{1}{e^{\frac{hc}{\lambda kT}}-1}.$$

Integrating this equation for λ=0 to *∞* gives $F_{i}$, the Fisher information for stellar centroid *i* and its reciprocal $\sigma_{\min}^{2}$, the minimum variance for the centroid position error in *x* direction, being *x* an axis perpendicular to the incoming light rays:(19)$$\frac{1}{\sigma_{\min}^{2}}=F_{i}=\int_{\lambda =0}^{\infty }\frac{dF_{i}}{d\lambda }d\lambda =12\zeta \left(5\right)\pi^{4}\cdot \frac{k^{5}}{h^{5}c^{4}}\cdot D^{4}t\cdot C_{i}T_{i}^{5},$$
where ζ(5)=1.0369277551… is the Riemann zeta function evaluated at 5. Since the aperture is symmetrical, Equation (Equation 19) gives the minimum centroiding variance for star *i* ($\sigma_{\min,i}^{2}$) along any axis perpendicular to the direction of incoming light rays. From this equation, it can be noted that the lower bound of the standard deviation on centroiding error along any axis perpendicular to the true direction of the star is proportional to $D^{-2}$ and $t^{-1/2}$, when the number of detected photons is sufficiently large. This means that the star sensor diameter has a much larger effect in the ultimate centroid accuracy and precision than the exposure time.

### 2.10. Estimating the Lower Bound of Attitude Error from Many Stars

This section follows the formulation given by Markley and Crassidis in Reference [30], Section 5.5. This formulation is valid when measurement errors are small, uncorrelated and axially symmetric around the true direction of stars, conditions fulfilled by our model, except for ideal star sensors with microscopic dimensions.

According to Equations (5.113) and (5.114) in Reference [30], the covariance matrix ($P_{\vartheta \vartheta }$) of the rotation vector error (δϑ) for an optimal attitude estimator is the inverse of the Fisher information matrix F:(20)$$P_{\vartheta \vartheta }=F^{-1},$$
with:(21)$$F=\sum_{i=1}^{N}\frac{1}{\sigma_{min,i}^{2}}\left[I_{3\times 3}-r_{i}^{\mathrm{true}}{\left(r_{i}^{\mathrm{true}}\right)}^{T}\right],$$
for the ideal star sensor model adopted in this work. In this equation, $I_{3\times 3}$ is a 3×3 identity matrix and $r_{i}^{\mathrm{true}}$ is the true direction of star *i*, represented by a unit vector expressed as a 3×1 column matrix. $r_{i}^{\mathrm{true}}$ is given in an inertial reference frame and *N* is the number of identified stars used in attitude computation.

### 2.11. A Compact Metric for the Attitude Error

Even though the covariance matrix $P_{\vartheta \vartheta }$ provides detailed information about the attitude uncertainty, as it has six independent parameters it has the disadvantage of being hard to visualize. Therefore, to perform comparisons, we use a more compact metric derived from it:(22)$${\left({\bar{\vartheta }}_{rms}\right)}^{2}=E\left\{\vartheta^{2}\right\}=\mathrm{tr}\left(P_{\vartheta \vartheta }\right).$$

The trace of the covariance matrix $P_{\vartheta \vartheta }$ gives the variance of the overall attitude error, that is, the sum of the variances of the attitude error around the three defining axes of the reference frame. It is also equal to the square of the limiting value of the root mean square (rms) of the angle theta (ϑ) of the Euler axis/angle parameterization of the attitude error when the number of attitude determinations tends to infinity.

When the star sensor diameter and exposure time are large enough so that most stars contributing to the Fisher information matrix *F* have many detected photons, the lower bound of the expected rms value of theta (${\bar{\vartheta }}_{rms,min}$) can be computed by Equations (Equation 19)–(Equation 22). These equations can also be rearranged in the following manner, which makes more explicit the dependence of ${\bar{\vartheta }}_{rms,min}$ with *D* and *t*:(23)$${\bar{\vartheta }}_{rms,min}=G\cdot D^{-2}\cdot t^{-1/2},$$
with
(24)$$G=\sqrt{\frac{h^{5}c^{4}}{12\zeta \left(5\right)\pi^{4}k^{5}}\cdot \mathrm{tr}\left[{\left(\sum_{i=1}^{N}C_{i}T_{i}^{5}\left[I_{3\times 3}-r_{i}^{\mathrm{true}}{\left(r_{i}^{\mathrm{true}}\right)}^{T}\right]\right)}^{-1}\right]},$$
*G* is a constant that depends only on stellar distribution around the star sensor, stellar brightness and on attenuation of stellar light by the intervening medium.

## 3. Discussion and Results

### 3.1. Star Catalogs Used

The Hipparcos star catalog was initially selected because it was, until very recently, one of the most accurate star catalogs available for precise attitude work. Therefore, we already had all the tools needed to process it. Unfortunately, the Hipparcos star catalog having less than 120,000 stars is too short to give an adequate basis for extrapolation. Therefore it was decided to include data from two larger catalogs, the Tycho-2 [31,32] with around 2.5 million stars and 2MASS [33] with about 470 million objects. A slightly more accurate basis for extrapolation would had been obtained if we had included larger star catalogs such as the Gaia DR2 star catalog [34] or the USNO-B star catalog [35]. However, this would have required us to rewrite most tools used in star catalog processing to make them run in a reasonable time in the hardware that was available to us.

The Hipparcos and Tycho-2 star catalogs give magnitudes in the optical regime (near ultraviolet, visible and near infrared), whereas the 2MASS star catalog gives magnitudes in the near/shortwave infrared bands *J*(1.25 μm), *H*(1.65 μm) and $K_{s}$(2.16 μm).

### 3.2. Adequacy of the Black-Body Approximation

In order to check the adequacy of the black-body approximation used in Section 2.4, we have performed a numerical integration of Equation (Equation 17) for some selected stars, using their actual spectra. It was observed that, given the color index used, the black-body approximation provides a good fit for some stars, but the fitting is not so good for all of them. Figure 6 compares the actual spectra of two stars with the spectra of their black-body equivalents, derived from their B−V color indexes and *V* magnitudes ($m_{V}$) given in Table 1 using the methods described in Section 2.5 and Section 2.6. Spectral fluxes in Figure 6 are given in power per unit area per frequency (or wavelength) decade.

Table 1 also presents a comparison between the lower bound of centroiding error obtained by numerical integration ($\sigma_{\min,num}$, in the last row of the table) and the lower bound of centroiding error $\sigma_{\min,BB}$ obtained from the black-body approximations. To show how $\sigma_{\min,BB}$ can vary depending on the spectral bands used for estimating the equivalent black-bodies, results are presented for two photometric systems—Johnson’s UBV and 2MASS JHK${}_{s}$, with the derived black-body parameters (*T* and *C*) also shown. As can be seen, the error in $\sigma_{\min,BB}$ is typically less than a factor of 2 but sometimes it can be much larger (see for example star VB8).

The magnitudes and color indexes listed in Table 1 were computed from spectra downloaded from the CALSPEC database (see Section 2.5). For the UBV system, the zero points that define the origin of the magnitude scales were computed using the method described in Section 2.5. For the 2MASS JHK${}_{s}$ system, the zero points were considered to be the zero-magnitude in-band fluxes listed on the third column of Table 2 from Cohen et al. [36].

#### Color Index Limiting Values

As explained in Section 2.5, the color indexes were limited to the interval that corresponds to a temperature range of 2000 K to 30,000 K. It was observed that, when the upper temperature limit was raised to more than 100,000 K, the Fisher information matrix would be dominated by a few very blue, hot stars where the interpolation from the color index curve versus temperature would give a very high temperature, much higher than their actual temperatures, leading to a significant underestimate of ${\bar{\vartheta }}_{rms,min}$. In fact, even the 30,000 kelvins upper limit adopted in this work might be too high, resulting that the ${\bar{\vartheta }}_{rms,min}$ estimated here is probably lower than the actual lower bound of attitude error attainable by star sensors.

The lower limiting temperature of 2000 K could perhaps be set to a lower value (e.g., 500 K), in order to better accommodate interstellar absorption and the existence of brown dwarfs. However, it was noted that this lower temperature limit has very little effect in the estimated value of ${\bar{\vartheta }}_{rms,min}$.

The optimal selection of temperature limits to be adopted for the black-body model will be a subject of a future work, if this model is not abandoned in favor of a more accurate stellar spectra model.

### 3.3. Results from Catalogs and Extrapolation

Some scripts were written to numerically evaluate the lower bound on star sensor attitude error for different star catalogs, different spectral bands and limiting the number of stars used in the computations to the *N* brightest cataloged stars, with *N* varying from two stars to the whole star catalog. Figure 7 shows results obtained with the catalogs described in Section 3.1 for D=1 m and t=1 s. The letter codes B-V, V-I, $B_{T}$-$V_{T}$, J-H, J-$K_{s}$ and H-$K_{s}$ indicate the spectral bands and catalog used for each curve. These curves form a basis for extrapolation (dashed lines) from where it is possible to obtain an interval that will very likely contain the true value of the parameter *G* in Equation (Equation 24) for our location in the galaxy. In this figure, we have chosen D=1 m and t=1 s instead of more typical star sensor values because the chosen values lead directly to the numerical value of *G* in SI units when performing the extrapolation.

Performing a rough extrapolation, we obtain for N = $300\cdot 10^{9}$ stars, an estimated number of stars in our galaxy [37], ${\bar{\vartheta }}_{rms}\approx$$1.2\cdot 10^{-14}\;$rad for the lower extrapolation curve and ${\bar{\vartheta }}_{rms}\approx 5.8\cdot 10^{-14}\;$rad for the upper extrapolation curve. However, there were many approximations made in the model, mainly the assumption of black-body spectra for stars. Therefore, the ${\bar{\vartheta }}_{rms}$ upper and lower estimates for D=1 m and t=1 s could still be wrong by a factor of 1.5 or 2. Hence, additional safety factors represented by the red vertical arrows along the $3\cdot 10^{11}$ stars dashed dotted line in Figure 7 were included. With these safety factors, and considering that the contribution of extragalactic sources is negligible (Section 3.3.1), which makes ${\bar{\vartheta }}_{rms}\left(N=300\cdot 10^{9}\right)\approx {\bar{\vartheta }}_{rms,min}$, it should be safe to assume that the true value of the *G* constant is: $7\cdot 10^{-15}$ rad m${}^{2}$ s${}^{1/2}<$*G*$<10^{-13}$ rad m${}^{2}$ s${}^{1/2}$.

#### 3.3.1. Contribution from Extragalactic Sources

The contribution of all existing extragalactic sources in the known Universe for the attitude accuracy is probably very small (probably less than 10% of the overall Fisher information). The reason for that is the vast distances between galaxies in comparison to their dimensions. For example, the nearest galaxy about the same size or larger than our galaxy is the Andromeda Galaxy. Its center lies about at a distance of 780 kpc from us [38], which is about 10–20 times the diameter of their disks.

Our galaxy, the Milky Way Galaxy, is orbited by many dwarf galaxies, such as the Small and Large Magellanic Clouds, but the total number of stars in these dwarf galaxies is less than 10% of the number of stars in our galaxy, therefore their contribution is also negligible.

Considering that the light intensity (and number of detected photons per unit time) falls off with the square of the distance from the source and that the Fisher information contributed by a star is proportional to the number of detected photons from that source, it is easy to see that the contribution from extragalactic sources will be small.

Nevertheless, some extragalactic sources were included in our estimates. Extragalactic sources can be divided in two categories, based on their apparent angular dimensions: extended sources and pointlike sources. Extended sources were excluded from analysis due to simplifying Assumption 9 in Section 2.2. However, many pointlike extragalactic sources were already present in the 2MASS Point Source Catalog which was used in Figure 7.

#### 3.3.2. Accuracy Degradation in Small Ideal Star Sensors with Short Exposure Times

When performing the extrapolation in Figure 7 to compute constant *G*, it was assumed that all point sources that can contribute to attitude knowledge would be observed. However, this is not always true, especially for very small star sensors, or when the exposure time *t* is very short. In these cases, many weak sources will have a low probability of being observed, due to the fact that for most measurements taken by the ideal star sensor no photon from these sources will hit its surface within the exposure time interval of *t*. This results in an additional degradation in the average accuracy of the star sensor, even when all assumptions described in Section 2.1 and Section 2.2 are fulfilled.

From Figure 7 it is possible to get a rough idea of how much the accuracy of an ideal star sensor suffers from this effect. This figure indicates that the accuracy is severely degraded if the average number of observed stars is less than $10^{4}$, but the degradation is small when, on average, more than $10^{8}$ stars are observed.

Numerical tests with sections of the 2MASS star catalog, containing stars from the middle sections and last sections when 2MASS was sorted by magnitude, indicate that when D=0.1 m and t=0.01 s, for more than half of the 2MASS catalog stars (more than $2.3\cdot 10^{8}$ stars), the expected number of photons arriving at the star sensor (ι) will be larger than one. Indeed, for stars in the middle of 2MASS, ι>1.6, meaning these will have a probability of being detected, from the Poisson distribution, larger than $1-e^{-1.6}=79.8\%$. From the distribution of stars versus limiting magnitude, we know that the average number of observed stars by an ideal star sensor (stars with at least one photon hitting the star sensor) will be roughly equal to the number of stars having ι>1.0. Therefore, it is safe to assume that for D⩾0.1 m and t⩾0.01 s much more than $10^{8}$ stars will be observed on average.

Considering that the number of photons from a given source detected by the star sensor is proportional to the product of its aperture area and exposure time, therefore, being also proportional to $D^{2}t$, it is easy to see that the same result will hold for any other combination of star sensor diameter *D* and exposure time *t* that satisfies the inequality $D^{2}t⩾0.0001$ m${}^{2}\cdot$s. Therefore, ideal star sensors that satisfy $D^{2}t⩾0.0001$ m${}^{2}\cdot$s will come close to the theoretical limit established by Equation (Equation 23), being probably within the uncertainty interval of constant *G* obtained from Figure 7.

The numerical tests described in this section can be performed by providing appropriate parameters to the function lower_bound_from_starcat.m in the “Appendix A”.

#### 3.3.3. Need to Consider Some Stars as Extended Sources

The lower bound on attitude uncertainty is so low that future star sensors would probably need to consider some stars as extended bodies and correct the effects of stellar spots (akin to sun spots, but in other stars) in their atmospheres to be able to come close to this theoretical lower bound, something that is unthinkable for current generation star sensors. For example, the star R Doradus—the star with largest apparent diameter after the Sun—has an apparent diameter of 57 ± 5 mas [(2.76 ± 0.25)$\cdot 10^{-7}$ rad] [39].

### 3.4. Effects of Field of View Restriction on Star Sensor Accuracy

Up to this point, we have considered unobstructed ideal star sensors, according to simplifying Assumption 17 in Section 2.2. However, in practice, real star sensors have limited FOVs (fields of view), and even if it were possible to design a star sensor with an unlimited field of view (FOV), parts of the sky would be obstructed by the spacecraft itself. Therefore, in this section we investigate how limitations in the FOV affect their accuracy.

The model adopted in this section is that of a diffraction limited and photon noise limited star sensor, based on the model presented in the previous sections, but with limitations in its FOV. Its FOV is considered to be a cone with a diameter varying from 360${}^{\circ }$ (full celestial sphere) down to 4${}^{\circ }$ (narrow FOV). For numerical computations, we assume an aperture diameter *D* of 10 cm and exposure time *t* of 10 ms. Computations were performed for two orthogonal pointing directions in the sky, labeled “target 1” and “target 2” and also for their combination. Due to practical reasons, we were forced to limit the catalog size used in this study (see Section 3.4.1). The estimates presented here were computed considering all stars in 2MASS up to J magnitude of 12.5750. This corresponds to the brightest 26,036,431 stars in J band, about 5% of the full 2MASS star catalog. For each star, equivalent black bodies have been computed from the J and H magnitudes. Results are presented in Table 2.

The first direction—labeled “target 1”—points towards the First Point of Aries (RA = 0 h, dec = 0${}^{\circ }$), which is a region with a low density of stars, whereas “target 2” points to (RA = 6 h, dec = 0${}^{\circ }$) in the Orion constellation, a region full of bright stars. The last column—labeled “both”—includes results that would be obtained when the individual measurements of identical single head star sensors pointing at “target 1” and “target 2” were merged using an efficient estimator. This is equivalent to having a star tracker with two optical heads.

As can be seen from Table 2, for a fixed aperture diameter *D* and exposure time *t*, there is a significant degradation in the attitude accuracy attainable by the star sensor as its FOV becomes narrower, even under the assumption that the star sensor is ideal, in the sense of being limited only by diffraction and photon noise. Also, as the FOV becomes narrower, the dependence on the region of the sky to where the star sensor is pointing becomes evident. We also observe, especially for narrow FOVs, that a significant gain in accuracy can be obtained when measurements from different directions orthogonal to each other are combined together. This is one of the reasons for many spacecraft having at least two star sensors (or star sensors with two or more optical heads) pointing at orthogonal or close to orthogonal directions.

The first row in Table 2 presents results for a star sensor having an unrestricted FOV of 4π sr. Given that the FOV in this case is the whole celestial sphere, naturally their accuracy will be the same, regardless of their pointing direction. By combining their measurements using an efficient estimator, but without using interferometry, the attitude uncertainty decreases by a factor of $\sqrt{2}$. Had these measurements been interferometrically combined, much larger gains in attitude accuracy would become possible. Notice that when this row was computed, we have ignored the fact that one star sensor will be partially obstructing the field of view of the other when they are placed close to each other.

It should be noted that the values presented in Table 2 have large uncertainties. The actual values of ${\bar{\vartheta }}_{rms}$ for FOV limited “ideal” star sensors with unlimited internal star catalog may be a factor of two or three times larger or smaller than the values shown in this table. Most of this uncertainty is caused by the approximation of real spectra of stars by the spectra of equivalent black-bodies (see Section 3.2) plus uncertainties in the cataloged magnitudes, followed by limitation in the size of the star catalog used when computing these estimates.

The results presented here can be better appreciated in Figure 8, in Section 3.5.

#### 3.4.1. Quality Assessment of Estimates Presented

Due to development and computational time constraints, we had to restrict the number of stars used in the study performed in this section. Out of the 20 sections generated by one of our preprocessing tools (file 2mass/sort_2mass_JHK.m in the Appendix A), we have used only the first section (generated file psc_JHK01.mat), containing the brightest stars in J band. Despite this limitation in this short study, the results are not very far from what would be obtained had we considered the full 2MASS star catalog. Judging from Figure 7, had we used all the stars in 2MASS, the estimated ${\bar{\vartheta }}_{rms}$ values would probably reduce by about 20% or 30%. This small improvement pales in comparison to the large uncertainty factor of two or three resulting from the approximation of real stellar spectra with spectra of equivalent black-bodies derived from cataloged magnitudes.

Another way to assess the quality of the estimates presented in Table 2 is to compare the values in the first row of Table 2 with the theoretical limit given in Section 3.3: For D=10cm and t=10ms we have from Equation (Equation 23) that $7\cdot 10^{-12}$ rad $<{\bar{\vartheta }}_{rms,min}<10^{-10}$ rad. Converting values to milli-arc-seconds, we find: 0.0014 mas $<{\bar{\vartheta }}_{rms,min}<0.0206$ mas. It can be seen that the tabulated value of ${\bar{\vartheta }}_{rms}=0.0073$ mas for a single star sensor with unobstructed FOV is well within the uncertainty range of ${\bar{\vartheta }}_{rms,min}$.

An ideal star sensor with D=10cm and t=10ms would also suffer a small loss in its accuracy due to the fact of rarely being able to observe extremely dim stars. However, as explained in Section 3.3.2, this effect is small for this combination of *D* and *t*, being smaller than the loss caused by limiting its internal catalog to stars with $m_{J}⩽12.575$.

### 3.5. Comparison with Existing Star Sensors

To give a feeling of how much room for improvement there is for future technology developments, Figure 8 compares the reported accuracy of ten different single head star sensors [40,41,42,43,44,45,46,47] with the theoretical lower bounds of equivalent spherical star sensors. The comparison is performed in terms of the combined metric $D^{2}t^{1/2}$, according to Equation (Equation 23), which makes it possible to compare many different star sensors and arrangements in a single plot.

In Figure 8, blue circles represent single star sensors used in isolation. The effective diameter used for computing the $D^{2}t^{1/2}$ metric in this arrangement has been taken as the approximate diameter of the smallest sphere that encloses its optical head, excluding its baffle. The second arrangement studied consists of two identical star sensors pointing at orthogonal directions used in combination, being represented by orange squares. The diameter used for computing the $D^{2}t^{1/2}$ metric for this arrangement has been taken as being 1.6 times larger than the previous. Even though for some star sensors it may be possible to get a tighter packing, we have chosen this factor of 1.6 to give some allowance for connectors and cables. For three out of ten star sensors, it was also possible to draw their positions in the plot considering their optical diameter aperture in place of the diameter of the minimum enclosing sphere. These are indicated by dark yellow markers to the left, connected by dashed light blue lines to their corresponding estimates described previously. Lines joining different arrangements of the same star sensors were added to facilitate comparisons. The figure also includes examples of “ideal”, but FOV limited and catalog limited star sensors, taken from Table 2. Their FOV diameter varies from $4^{\circ }$ to $360^{\circ }$. The results obtained for two “ideal” star sensors in combination (last column of Table 2) are labeled “ideal, two optical heads” in the figure. The diameter of the smallest sphere enclosing two “ideal” star sensors is twice the diameter of a single “ideal” star sensor.

The solid line at the bottom left part of this plot denotes the lower estimate of the lower bound of the attitude error ${\bar{\vartheta }}_{rms,min}$, derived from the lower curve in Figure 7. The dashed line immediately above this solid line is the upper estimate of the the lower bound of the attitude error ${\bar{\vartheta }}_{rms,min}$, derived from the upper curve in Figure 7. The true lower bound of attitude error (${\bar{\vartheta }}_{rms,min,true}$) that can be obtained using solely electromagnetic radiation emitted by bodies outside the Solar System should lie between these two curves. No star sensor that satisfies the Assumptions 4 and 8 stated in Section 2.1 should be able to surpass ${\bar{\vartheta }}_{rms,min,true}$ without making use of additional attitude reference sources, such as the Sun and other Solar System objects (excluded from analysis by the simplifying Assumption 12 in Section 2.2) or artificial attitude references.

We have opted to exclude the baffle in the estimate of *D*, because the sole reason for including a baffle in star sensors is to protect them from being temporarily blinded by stray light coming from the Sun and other bright sources (Earth, Moon and other spacecraft parts), something that would not happen in the absence of Solar System objects. Also, including the baffle would greatly inflate the diameter *D* of the smallest enclosing sphere. The accuracy used in this plot for star sensors used in isolation was derived from the reported (1-σ) noise equivalent angle or attitude accuracy, using the equation $\vartheta_{rms}^{2}=\sigma_{x}^{2}+\sigma_{y}^{2}+\sigma_{z}^{2}$, being $\sigma_{x}=\sigma_{y}$ the uncertainties around the cross-boresight axes (pitch and yaw angles) and $\sigma_{z}$ the uncertainty around the boresight axis (roll angle). For details on how the accuracy of two star sensors used in combination was computed, see file plotFig8.m in the Appendix A.

It is interesting to note that for most star sensor examples plotted in Figure 8, a combination of two star sensors comes closer to the theoretical limit than a single star sensor, even with the penalty of ${\left(1.6\right)}^{2}=2.56$ in the plotted $D^{2}t^{1/2}$ metric. This happens because in single headed star sensors, the uncertainty in the roll angle is usually much larger than those in the yaw and pitch angles, but when measurements from two star sensors pointing at orthogonal or almost orthogonal directions are merged, it is possible to compensate the poor roll angle measurement of one unit with one of the more accurate yaw/pitch measurements from the other unit (or their combination, depending on how the defining axes are oriented), thus allowing accurate attitude knowledge in all three axes. The explanation for the poor accuracy in roll angle lies in the following fact: A rotation around the boresight axis (roll) by a small angle (for example, one degree) will cause a much smaller shift in the scene that a single head star sensor is observing than a rotation by the same angle around a cross-boresight axis (yaw/pitch or a combination of them). A rotation around the boresight axis will be perceived by the star sensor as a rotation of stars around a point in the center of its FOV, whereas for a rotation around a cross-boresight axis the star sensor will see all stars moving in the same direction across its FOV.

It should be noted that the positions of the star sensors listed in this plot may have considerable errors. For example, for some of them, the actual $D^{2}t^{1/2}$ values and rms attitude errors could be actually a factor of two or three smaller than those shown in the plot. Reasons for these discrepancies lie in the fact that many star sensor manufacturers, being conservative, quote worst case conditions in their product briefs, plus the many approximations and assumptions we have made when creating this plot. Therefore, we advise against the idea of trying to compare different products based on Figure 8, considering that it was created only from publicly available information, not always complete or accurate. Detailed information on the sources and the methods used to create this plot can be seen in the comments inside plotFig8.m which is included in the supplementary materials.

As can be seen, there is still a lot of room for improvement in star sensors. The theoretical lower bound is around six to eight orders of magnitude lower than what is currently attained by most star sensors.

## 4. Conclusions

To our best knowledge, Chapter 7 in the doctoral thesis of the first author [14] was the first work which provided estimates on the lower bound of attitude uncertainty attainable by star sensors in the Solar System’s stellar neighborhood. These derived limits are valid in our stellar neighborhood regardless of the star sensor technology, when no Solar System body can be used to significantly augment attitude determination (for example, for a spacecraft a few light years from home in the far future). Being the first work to attempt to derive the numerical values of this limit, it did not aim for much accuracy. This explains the large factor (of about one order of magnitude) between the upper and lower estimates of this lower bound on attitude uncertainty presented in that work and revised here. Nevertheless, these results suffice for the purpose of obtaining an order of magnitude evaluation on how much room for improvement there exists for current state of the art star sensors. It is shown that the accuracy of current star sensors can still improve by about seven orders of magnitude before reaching the ultimate limits imposed by laws of Physics and stellar distribution in our stellar neighborhood.

To facilitate verification of the results presented in this work and to foster a culture of open collaboration in the scientific community, the authors make the source code of the routines used to generate results presented in this work available to anyone interested as a free open source software, accompanying the on-line version of this work. To assure long term preservation, these routines were also published in other places, described below in the Section “Appendix A.”

## Figures and Tables

**Figure 1 sensors-19-05355-f001:**
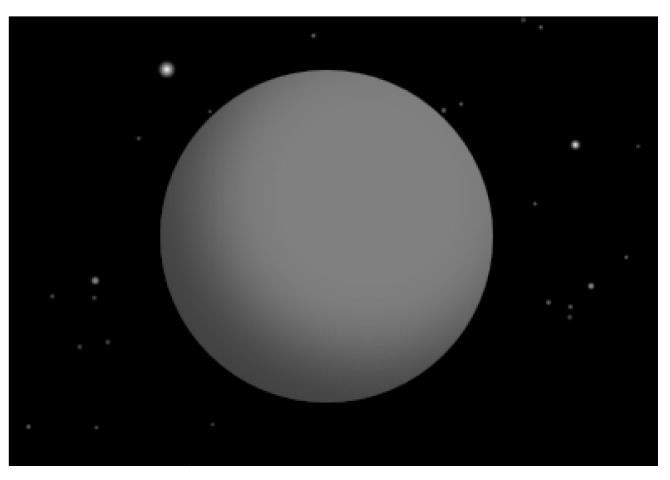
Ideal star sensor model with stars on the background. To aid visualization, unrealistically represented by a gray sphere here.

**Figure 2 sensors-19-05355-f002:**
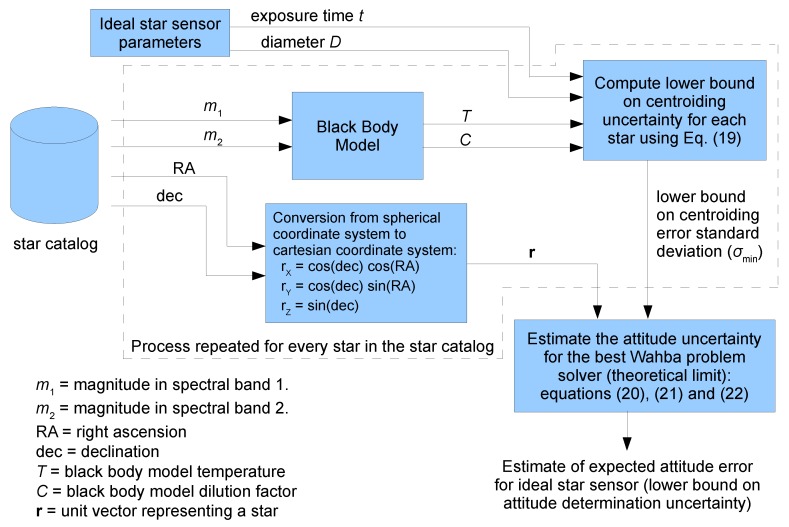
Model for estimating the theoretical lower bound on attitude uncertainty for star sensors.

**Figure 3 sensors-19-05355-f003:**
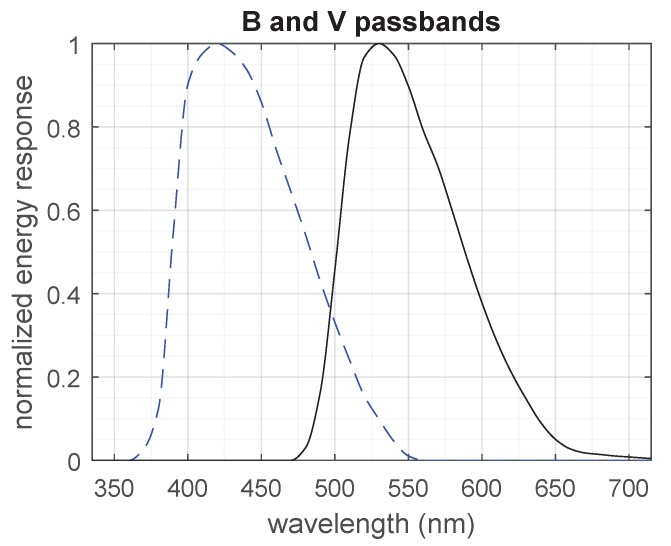
Spectral energy response of the *B* (blue) and *V* (visual) bands. The *B* band is the blue dashed curve to the left.

**Figure 4 sensors-19-05355-f004:**
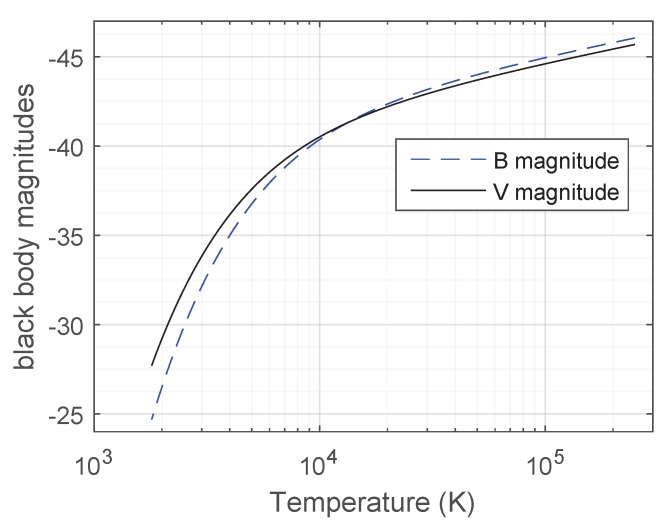
Apparent magnitudes of a black body for an observer lying on its surface and looking down towards its center versus black body temperature, in the Johnson-Morgan *B* and *V* bands. Note that the vertical axis of this plot is reversed, with more negative magnitudes (brighter black-bodies) at the top.

**Figure 5 sensors-19-05355-f005:**
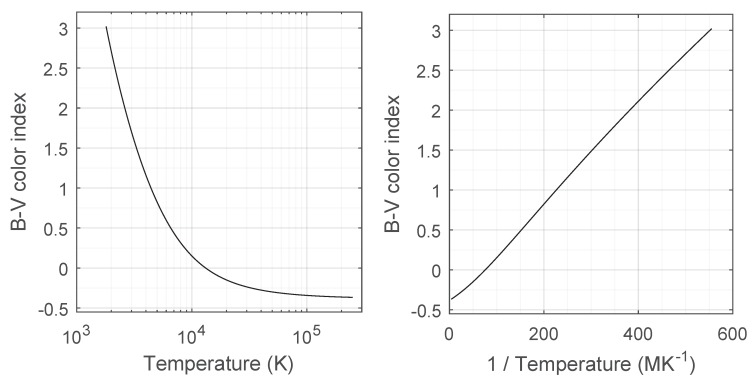
Relation between temperature (or its reciprocal) with B−V color index for black bodies.

**Figure 6 sensors-19-05355-f006:**
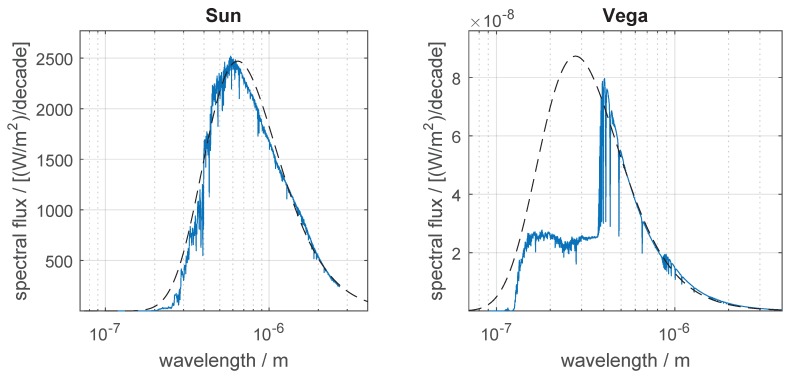
Comparison between the actual spectra for the Sun and Vega (α-Lyr) with the spectra of their black-body equivalents derived from their B−V color indexes and *V* magnitudes with the methodology explained in Section 2.5 and Section 2.6. Actual spectra represented by continuous line. Dashed lines represent the spectra of equivalent black-bodies.

**Figure 7 sensors-19-05355-f007:**
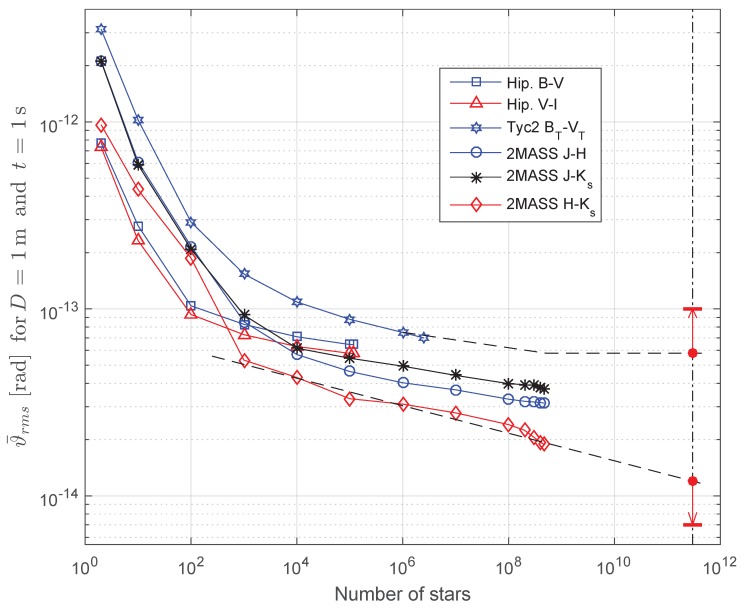
Estimates of ${\bar{\vartheta }}_{rms}$ for an ideal star sensor with D=1 m and t=1 s, obtained from the following star catalogs: Hipparcos, Tycho-2 and 2MASS and their subsets of brightest stars. Extrapolation curves shown as dashed lines.

**Figure 8 sensors-19-05355-f008:**
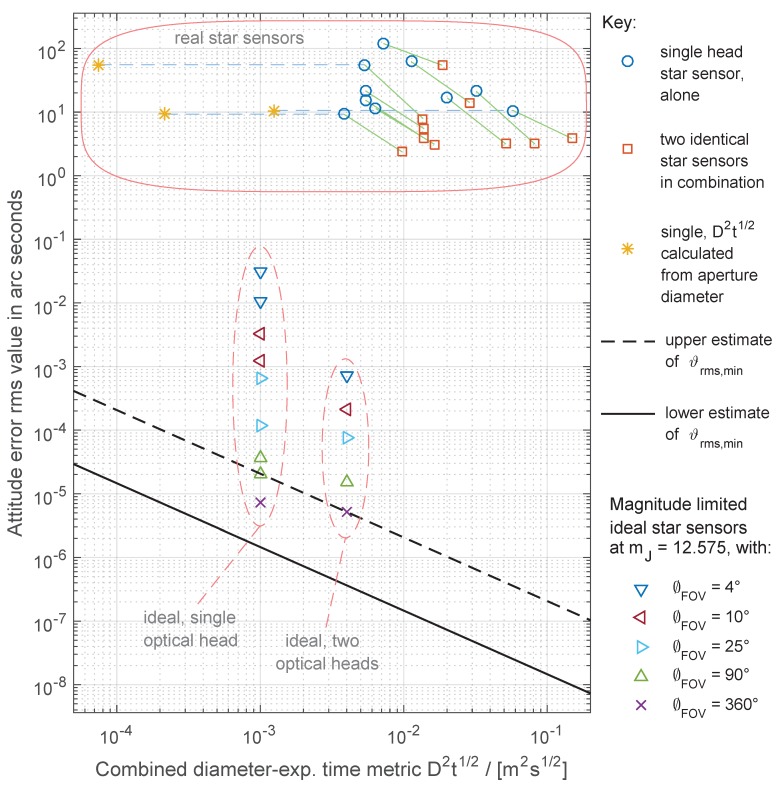
Comparison between some commercial star sensors with the upper and lower limits of the theoretical lower bound on attitude uncertainty.

**Table 1 sensors-19-05355-t001:** Comparison for some selected stars when D=10cm and t=100ms.

Parameter			Star				
Name/Symbol	Unit		Vega	1757132	Sun *	KF06T2	VB8
spectral type	-		A0V	A3V	G2V	K1.5III	M7V
$m_{V}$	mag		+0.030	11.81	−26.75	13.97	16.80
B−V	mag		−0.001	00.26	+00.65	01.18	02.01
*T*	K		13,231	8580	005711	3951	2613
*C*	1		$2.96\cdot 10^{-17}$	$1.87\cdot 10^{-21}$	2.42·10−5*	$9.59\cdot 10^{-21}$	$2.01\cdot 10^{-20}$
$\sigma_{\min,BB}$	rad		$1.19\cdot 10^{-10}$	4.42·10−8*	$1.08\cdot 10^{-15}$	1.36·10−7*	2.63·10−7*
$m_{H}$	mag		−0.004	11.23	−28.24	11.26	09.17
$H-K_{s}$	mag		−0.005	00.02	+00.04	00.10	00.34
*T*	K		10,417	8961	008059	6050	3282
*C*	1		$4.80\cdot 10^{-17}$	$1.94\cdot 10^{-21}$	1.41·10−5*	$3.72\cdot 10^{-21}$	$1.05\cdot 10^{-19}$
$\sigma_{\min,BB}$	rad		$1.70\cdot 10^{-10}$	3.89·10−8*	$5.94\cdot 10^{-16}$	7.50·10−8*	6.50·10−8*
$\sigma_{\min,num}$	rad		$1.74\cdot 10^{-10}$	5.53·10−8*	$1.14\cdot 10^{-15}$	1.48·10−7*	9.24·10−8*

*  Not used in the results presented in this work, due to its extreme proximity (see explanation at the end of Section 2.2).

**Table 2 sensors-19-05355-t002:** Estimated accuracies for an “ideal” field of view (FOV) limited star sensor versus its FOV diameter, for D=10cm and t=10ms. **Note:** Estimates have large uncertainties, actual values may be up to three times as larger or smaller than the values shown.

FOV	Attitude Error (${\bar{\vartheta }}_{\mathrm{rms}}$)		
	(in Milli-Arc-Seconds)		
	Target 1	Target 2	Both
360${}^{\circ }$	0.0073	0.0073	0.0051
90${}^{\circ }$	0.0369	0.0202	0.0152
25${}^{\circ }$	0.641	0.120	0.0754
10${}^{\circ }$	3.29	1.231	0.213
4${}^{\circ }$	31.6	10.8	0.720

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
