# Peer review of "Theoretical Limits of Star Sensor Accuracy†"

_sensors, 2019, doi:10.3390/s19245355_

Round 1

Reviewer 1 Report

The paper provides a way to calculate the fundamental lower bound of star sensors’ measurement accuracy using Heisenberg’s uncertainty principle and analyses the accuracy limit for different stars and different star numbers. The  estimation of the lower bound is not only meaningful with star sensors, but also can be used in other optical measurement systems. Here are some suggestions below.

In my opinion, the logic of the abstract is not clear. The trend of miniaturizing spacecraft instruments and the fundamental limit on star sensor accuracy are two different concepts. The organization could be improved. The key words should be modified. “star sensors”, “star trackers”, “attitude sensors” actually are the same instrument here. Since the author focuses on the accuracy of stars sensors, in my opinion, the Table 1 in the results could be better if it is for star sensors (For example, D is 10 millimeters, and t is 100 milliseconds. Use parameters of star sensors instead of telescopes).

Author Response

First, we would like to say thanks for you giving us indications of points that could be improved in our work! We really appreciate all your comments!

Our replies are in the attached file.

If it does not open, here is a backup copy:

---------------------------------------------------------------------

Responses to Reviewer 1 Comments:

 (reviewer 1 comments from 29 Oct 2019 17:11:35)

1 - The estimation of the lower bound is not only meaningful with star sensors, but also can be used in other optical measurement systems

Response 1: Thanks for this comment! Indeed, the limits described in our work are applicable to any optical measurement system in the situation where no solar system body can be used to augment the attitude knowledge. We only have refrained from stating that these limits can be used in other optical measurement systems because we have not yet considered the Sun and other solar system objects in our computations, but we have plans in doing so in the future.

2 - In my opinion, the logic of the abstract is not clear. The trend of miniaturizing spacecraft instruments and the fundamental limit on star sensor accuracy are two different concepts.

Response 2: This is true. However, even though these are two different concepts, the trend of miniaturizing spacecraft instruments provides the motivation for the study. The smaller and more capable are future star trackers, the closer they will be to the fundamental limits derived in our work.

3 - The key words should be modified. “star sensors”, “star trackers”, “attitude sensors” actually are the same instrument here.

Response 3: Following your suggestion, we have removed the keywords “star trackers” and “attitude sensors” from the list of keywords. We also have uniformized the usage of terms “star tracker” to “star sensor.”

4 - Since the author focuses on the accuracy of stars sensors, in my opinion, the Table 1 in the results could be better if it is for star sensors (For example, D is 10 millimeters, and t is 100 milliseconds. Use parameters of star sensors instead of telescopes).

Response 4: We have replaced the values in this table, using parameters more typical of a star sensor: D = 10 cm and t = 100 ms.

Reviewer 2 Report

This paper is very interesting, and I enjoyed reviewing it. I have a single major concern.

The idea of considering an ideal spherical sensor for the derivation of the star sensor accuracy limit is not fully convincing. Even if such a sensor could be realized, and all the other assumptions on its operation were met (e.g., assumptions 3, 4 and 5 at page 2), I do not see how it could deal with the occlusions limiting the FOV which can be actually exploited (thus, of course, the accuracy would not be independent of the sensor pointing). The main sources for occlusions could be (1) the structure of the satellite on which the star sensor is installed, (2) the Earth (which would occupy most of the FOV, especially in LEO), (3) the Sun (even if not considered in the estimation it would occupy a not negligible portion of the FOV). I think this point is very important because, despite the goal of the paper is to derive the accuracy of a theoretical sensor, the physical limits associated to its installation should be considered.

Minor comment

Could you clarify which assumption is adopted regarding the resolution of the sensor? Are you assuming to have an unbounded capability to distinguish between stars which have very low angular separation on the celestial sphere?

Author Response

Dear Reviewer 2, we would like to say thanks for all your comments, which surely will help us to improve the quality of our work. Thanks!

We have written our comments in the attached file. Please try to open it, as it has the proper formatting. If by some reason it does not open, here follows a backup copy:

------------------------------------------------------------------------------

Response to Reviewer 2 comments:

(Reviewer 2 comments from 03 Nov 2019 16:14:56)

1 [...] I do not see how it could deal with the occlusions limiting the FOV [...] The main sources for occlusions could be (1) the structure of the satellite [...], (2) the Earth [...], (3) the Sun [...]. I think this point is very important because, despite the goal of the paper is to derive the accuracy of a theoretical sensor, the physical limits associated to its installation should be considered.

Response 1:

Dear Reviewer 2. Thanks for pointing out this issue!

We have not considered in the study the physical limits associated with the star sensor installation, because doing so would be impractical, considering that these conditions vary widely between missions and particular circumstances within a mission (e.g., if the spacecraft is near the Earth or far from the Earth, and which stars are being obstructed by the Earth).

Nevertheless, to make matters clear, we have added the simplifying assumption 9:

there is no obstruction from spacecraft structures or nearby bodies.

and the following paragraph in section 2.2:

Regarding simplifying assumption 9, had there been any obstruction in the field of view of the ideal star sensor (such as obstruction from nearby bodies or obstruction by spacecraft structures), its accuracy would necessarily be worse, since the obstructed stars would no longer contribute to the attitude information gathered by the star sensor. Therefore, we assume that it has unrestricted view of the whole celestial sphere.

2 -  ...occlusions limiting the FOV which can be actually exploited...

Response 2:

Even though occlusions could perhaps be exploited as additional attitude information sources in some setups, as noted by the reviewer (e.g., by taking note of which stars are obstructed and which are visible), I’m not sure that this information would be able to compensate for the lost information from obstructed stars, probably not. This idea could perhaps be useful in a scenario where individual star directions are measured with an error much, much larger than the theoretical limits imposed by diffraction and photon noise, which is not the case considered in our work.

Our understanding is that any occlusion limiting the FOV would be detrimental to the accuracy of the star sensor, especially for an ideal star sensor which is diffraction and photon noise limited (such star sensor still does not exist). By limiting the number of stars that are available, we are limiting the number of attitude sources that can be used to compute an optimal estimate of the star sensor attitude (in relation to the inertial space). In fact, any photon of stellar origin that is not detected by the star sensor is an information lost, which could otherwise contribute to its attitude knowledge. So, to give an answer to your comments, in practice, the accuracy of an ideal spherical star sensor mounted in a real spacecraft would be further limited by obstruction by spacecraft structures and nearby bodies (such as Earth). For real star sensor, they are already severely limited by their restricted field of views. No wonder their accuracy is many orders of magnitude worse than the theoretical limits discussed in our work.

3 - Could you clarify which assumption is adopted regarding the resolution of the sensor? Are you assuming to have an unbounded capability to distinguish between stars which have very low angular separation on the celestial sphere?

Response 3:

The difference in the results if we consider or not an unbounded capability to distinguish between stars that are very close to each other in the celestial sphere is negligible, since considering them as separate stars or a single star with the combined brightness when computing the Fisher information matrix (Eq. 21) will give for all practical purposes the same results when the angular separations between combined stars is very small. This can be proven by considering what happens in Eq. 21 when we take two stars very close to each other and approximate their unit vectors ri to the same value. In that case their inverse variances (1/σi) will add together resulting in the variance of an equivalent star with the combined brightness of the stars that were merged.

Reviewer 3 Report

The authors of the article prove the existence of the theoretical accuracy limit of star trackers. To do this, they introduced and reviewed the ideal spherically symmetric star tracker. They received the following conclusions: there is a theoretical accuracy limit for stellar trackers, it is determined by quantum uncertainty in photon registration, the theoretical limit is 6-8 orders of magnitude less than the errors of modern sensors (Figure 8).

The presentation of many questions in the article is excessively detailed. This is probably due to the fact that this article is a revised version of the PhD-thesis chapter of one of the co-authors. I think that the authors could reduce the text of the article without any influence on their conclusions. However, if the authors wish and with the consent of the editorial board of the journal, the presentation style of the article can be left unchanged.

An article may be published after the following comments have been corrected.

Line 70. At the end of the line should be a period, not a semicolon. The numbering of the assumptions in subsections 2.1 and 2.2 is best made continuous or in line 112 indicate that 6 is a simplifying Figure 5. If the B-V dependence is close to 1/T, then the curve in the left graph should look like a straight line, since in this graph the logarithmic scale for temperature. Lines 315-316. Either remove the value of 1mm, or justify this value. Equation (21), lines 319-326. Equation (21) is taken from [30]. The quantity \sigma_i appears in this equation, which is also defined in [30]. Apparently, the definition of \sigma_i differs from that used in this article, since three pages in the appendix are devoted to the proof of the equality
\sigma_i = \sigma_{\ min, i}. The difference in the definition of \sigma_i from [30] is not given in this article. I propose to replace \sigma_i with \sigma_{\min, i} in equation (21), and remove paragraph 324-326 and Application. Lines 322-323, footnote 8. According to assumption 6 of section 2.1 and 6 of section 2.2, the reference system of the star tracker coincides with the inertial reference system. Therefore, the phrase “some reference frame R” on line 322 and footnote 8 must be deleted. Section 3.3. For estimates, the value D = 1 m was used. Star sensors of this diameter are not made. It is better to use D = 1 cm or 10 cm. Figure 7. The value of “trace(covariance matrix)” is unphysical and not descriptive. Instead, necessary to give a graph for rms error. Figure 7. Present day the most accurate astrometric catalog is Gaia DR2. Why it not used in Figure 7? Line 399. Write “300 10^9” instead of “300 billion” and remove footnote 11. Line 409. According to equation (24), subject to the assumptions of sections 2.1 and 2.2, the coefficient G is independent of D and t. In addition, it is not clear where the values of D> 0.1m and t> 0.01s are taken from. Section 3.2.2. When considering extragalactic sources, their nonzero sizes must be discussed. Line 448. Where do the values D> 0.1m and t> 0.01s come from? Lines 480-483. Using the bodies of the solar system as additional reference sources violates assumptions 6 from section 2.1 and 5, 8 from section 2.2. Figure 8. It is necessary to give a table with the characteristics of the star trackers used in Figure 8 and bibliographic references to their sources. Paragraph 458- ... The discussion about baffles looks strange. In the public domain there is information about a large number of models of star trackers, including focal lengths and the diameters of their lenses. Lines 481- See comment 9. Section 4. Since the obtained theoretical limit of accuracy is 6–8 orders of magnitude less than that achieved, the continuation of this work does not make sense in the next hundred years. Section “Future work” I propose to delete. Lines 524-529. I propose to transfer this text to the description of the supplementary materials. Lines 530-531. Remove. Appendix. Remove, see comment 5.

Author Response

Dear reviewer 3.

First of all, thanks for your valuable comments! We are really grateful for all your comments and contributions to the quality of our work. Thanks!

Our answers are in the attached file. Please try to open it before reading our answers here. If it does not open by some reason, we pasted its contents here as a backup:

-------------------------------------------------------------------------------

Responses to Reviewer 3 comments:

 (Reviewer 3 comments from 30 Oct 2019 11:47:16)

Notice: line numbers refer to the version of our paper without annotations.

A version with annotations is also available.

1 - Line 70. At the end of the line should be a period, not a semicolon.

Response: Error corrected.

2 - The numbering of the assumptions in subsections 2.1 and 2.2 is best made continuous or in line 112 indicate that 6 is a simplifying assumption.

Response: We added the word “simplifying” to make it clear that we are referring to the simplifying assumption 6 in section 2.2. Thanks for spotting this error!

3 - Figure 5. If the B-V dependence is close to 1/T, then the curve in the left graph should look like a straight line, since in this graph the logarithmic scale for temperature.

Response:

Not really, because the vertical scale of the left plot is linear on B-V, not logarithmic. Had the vertical scale also be logarithmic on B-V, and the b offset (defined below) be zero, then the left graph would be similar to a straight line.

Proof: Suppose that the right plot is a straight line, then we would have: B-V = a/T + b, with a and b constants. Let x be the horizontal coordinate (in pixels or in cm) in the left plot, then T = T0*exp(c*x), with T0 and c being constants depending on the horizontal scale of this graph. Substituting this last equation into the former, we obtain: B-V = (a/T0)*exp(-cx) + b.

Comment: Perhaps this confusion originated from the fact that color indexes are differences between magnitudes (measured in different spectral bands) and magnitudes themselves are the logarithm of stellar brightness, meaning that the plot to the left is actually a log-log plot on the ratio of signals between bands B and V ( flux_in_B / flux_in_V ) versus temperature. In fact, we have B-V = -2.5 * log10(constant * flux_in_B / flux_in_V).

4 - Lines 315-316. Either remove the value of 1mm, or justify this value.

Response: Replaced “ideal star trackers with very small diameters, much less than 1 mm” with “ideal star trackers with microscopic dimensions.

5 - Equation (21), lines 319-326. Equation (21) is taken from [30]. The quantity σi appears in this equation, which is also defined in [30]. Apparently, the definition of σi differs from that used in this article, since three pages in the appendix are devoted to the proof of the equality σi = σmin, i. The difference in the definition of σi from [30] is not given in this article. I propose to replace σi with σmin, i in equation (21), and remove paragraph 324-326 and Application.

Response: The modifications have been performed as suggested by the reviewer. Appendix A and paragraph 324-326 have been deleted.

6 - Lines 322-323, footnote 8. According to assumption 6 of section 2.1 and 6 of section 2.2, the reference system of the star tracker coincides with the line 322 and footnote 8 must be deleted.

Response:

Line 322 (now 328) has been changed to:

       ritrue is given in an inertial reference frame and N is the number of ...

and footnote 8 has been deleted.

7 - Section 3.3. For estimates, the value D = 1 m was used. Star sensors of this diameter are not made. It is better to use D = 1 cm or 10 cm.

Response: We have replaced the values in this table, using parameters more typical of a star sensor: D = 10 cm and t = 100 ms.

8 - Figure 7. The value of “trace(covariance matrix)” is unphysical and not descriptive. Instead, necessary to give a graph for rms error.

Response: We have replaced “trace(covariance matrix)” with the expected rms error in the graph.

9 - Figure 7. Present day the most accurate astrometric catalog is Gaia DR2. Why it not used in Figure 7?

Response:

The authors are aware that Gaia DR2 would benefit the paper. However, doing this would require a time that is way longer than the 10 days suggested by MDPI for providing the revised version. We estimate that the time required for this task would be at least a month.

We have added the following text in the first paragraph of Section 3.1:

A slightly more accurate basis for extrapolation would had been obtained if we had included larger star catalogs such as the Gaia DR2 star catalog[34] or the USNO-B star catalog[35]. However this would have required us to rewrite most tools used in star catalog processing to make them run in a reasonable time in the hardware which was available to us.

10 - Line 399. Write “300 10^9” instead of “300 billion” and remove footnote 11.

Response: The modifications have been performed as suggested by the reviewer.

11 - Line 409. According to equation (24), subject to the assumptions of sections 2.1 and 2.2, the coefficient G is independent of D and t. In addition, it is not clear where the values of D> 0.1m and t> 0.01s are taken from.

Response: If the condition (D> 0.1m and t> 0.01s) is not fulfilled, there may be cases where the number of observed stars would be much less than 8·108, which is the number of stars where the upper extrapolation curve becomes flat in Figure 7. This is explained in detail in Section 3.3.1.

12 - Section 3.2.2. When considering extragalactic sources, their nonzero sizes must be discussed.

Response:

We can divide extragalactic sources into two categories:

- extended sources;

- point-like sources;

Extended sources, such as nearby galaxies were not considered in the analysis due to simplifying assumption 1 (see lines 106-108).

Some point-like extragalactic sources (such as quasars and distant galaxies) are already being considered, since many of them are included in the 2MASS Point Source Catalog which was used in this work (see lines 108-111 in our work).

The following paragraph has been included in Section 3.2.2 (lines 438-442):

Nevertheless, some extragalactic sources were included in our estimates. Extragalactic sources can be divided in two categories, based on their apparent angular dimensions: extended sources and point-like sources. Extended sources were excluded from analysis due to simplifying assumption 1 in Section 2.2. However, many point-like extragalactic sources were already present in the 2MASS Point Source Catalog which was used in Figure 7.

13 - Line 448. Where do the values D> 0.1m and t> 0.01s come from?

Response: They come from Section 3.3.1. See also our reply to your question 11.

14 - Lines 480-483. Using the bodies of the solar system as additional reference sources violates assumptions 6 from section 2.1 and 5, 8 from section 2.2.

Response: Section 4 – “Future work,” which did include this text (lines 497-500), has been deleted.

15 - Figure 8. It is necessary to give a table with the characteristics of the star trackers used in Figure 8 and bibliographic references to their sources.

Response: We have updated Figure 8. Since the table with characteristics of the star trackers used in Figure 8 would be very large, it was not included in the paper, but this information, including the methods used to derive it, can be found in file plotFig8.m, which is included in the supplementary material.

16 - Paragraph 458- ... The discussion about baffles looks strange. In the public domain there is information about a large number of models of star trackers, including focal lengths and the diameters of their lenses.

Response: We have rewritten most of this and the following paragraph using new data included in the script (plotFig8.m) used to generate Figure 8. 

17 - Lines 481- See comment 9.

Response: Section 4 – “Future work” has been deleted.

18 - Section 4. Since the obtained theoretical limit of accuracy is 6–8 orders of magnitude less than that achieved, the continuation of this work does not make sense in the next hundred years. Section “Future work” I propose to delete.

Response: Suggestion accepted. Section “Future work” has been deleted.

19 - Lines 524-529. I propose to transfer this text to the description of the supplementary materials.

Response: Following your suggestion, we will include this text in the description of the supplementary materials. However we plan to keep a copy of this text in the article. Reason: By keeping this information in the article we assure that our readers will be able to download the supplementary material from other sources and check its integrity even if MDPI ceases to exist.

20 -  Lines 530-531. Remove.

Response: All articles published in Sensors include an “Author Contributions” section.

21 - Appendix. Remove, see comment 5.

Response: The appendix has been removed.

Round 2

Reviewer 2 Report

I thank you for addressing my major comment.

However, I must say that I am not fully satisfied by the answer.

Indeed, it is clear that if the FOV is obstructed or if only a portion of the FOV is considered, the ideal sensor would provide a worse attitude accuracy than the one obtained in the manuscript (Figure 8). It is also clear that such lower accuracy would depend on the pointing as well as on any other mission-related circumstance.

However, I am convinced that it would be valuable for readers to understand the “weight” of such an assumption.

Which would the accuracy level be if a limited FOV was considered? Where would the accuracy of such ideal sensor be placed in Figure 8?

Considering the obvious variability of such accuracy from factors like the sensor pointing in the celestial sphere, a single example, though not fully explicative, would still be enough to provide the reader with an idea about the achievable accuracy level with respect to state-of-the-art star sensors.

Author Response

Dear reviewer.

Please see the attached file for our responses.

If it does not work, we have copied the text here a backup:

------------------------------------------------------------

Dear Reviewer 2. First of all, we would like to say thanks for all your comments, which surely will help us to improve the quality of our work. Thanks! 

Response to Reviewer 2 comments:

(Reviewer 2 comments from 17 Nov 2019 16:14:56)

1 Which would the accuracy level be if a limited FOV was considered? Where would the accuracy of such ideal sensor be placed in Figure 8?

Considering the obvious variability of such accuracy from factors like the sensor pointing in the celestial sphere, a single example, though not fully explicative, would still be enough to provide the reader with an idea about the achievable accuracy level with respect to state-of-the-art star sensors.

Response 1: Dear reviewer 2.

We have created a new section, Section 3.4, comparing the performance of limited field of view “ideal” star sensors. In this section, there is Table 2 which gives predicted performances depending on the field of view of the star sensor and the direction in the sky the star sensor is pointing to.

We also include estimates for combinations of two star sensors, pointing at orthogonal directions. A large improvement in attitude accuracy is obtained.

Figure 8 has also been updated with new data from Table 2.

In the supplementary materials, a new script (calcTable2.m) has been created to compute these estimates.
